# Sintering Behaviors, Microstructure, and Microwave Dielectric Properties of CaTiO_3_–LaAlO_3_ Ceramics Using CuO/B_2_O_3_ Additions

**DOI:** 10.3390/ma12244187

**Published:** 2019-12-13

**Authors:** Min-Hang Weng, Chihng-Tsung Liauh, Shueei-Muh Lin, Hung-Hsiang Wang, Ru-Yuan Yang

**Affiliations:** 1School of Information Engineering, Putian University, Putian 351100, China; hcwweng@gmail.com; 2Institute of Mechanical Engineering, Kun Shan University, Tainan 710, Taiwan; liauhct@mail.ksu.edu.tw (C.-T.L.); smlin@mail.ksu.edu.tw (S.-M.L.); ashon0659@yahoo.com.tw (H.-H.W.); 3Green Energy Technology Center, Kun Shan University, Tainan 710, Taiwan; 4Graduate Institute of Materials Engineering, National Pingtung University of Science and Technology, Pingtung County 912, Taiwan

**Keywords:** microwave, ceramics, density, dielectric properties

## Abstract

The effect of CuO/B_2_O_3_ additions on the sintering behaviors, microstructures, and microwave dielectric properties of 0.95LaAlO_3_–0.05CaTiO_3_ ceramics is investigated. It is found that the sintering temperatures are lowered efficiently from 1600 °C to 1350 °C, as 1 wt % CuO, 1 wt % B_2_O_3_, and 0.5 wt % CuO +0.5 wt % B_2_O_3_ are used as the sintering aids due to the appearance of the liquid phase sintering. The microwave dielectric properties of 0.95LaAlO_3_–0.05CaTiO_3_ ceramics with the sintering aid additions are strongly related to the densification and the microstructure of the sintered ceramics. At the sintering temperature of 1300 °C, the 0.95LaAlO_3_–0.05CaTiO_3_ ceramic with 0.5 wt % CuO + 0.5 wt % B_2_O_3_ addition shows the best dielectric properties, including a dielectric constant (ε_r_) of 21, approximate quality factor (Q × f) of 22,500 GHz, and a temperature coefficient of the resonant frequency (*τ_f_*) of −3 ppm/°C.

## 1. Introduction

Microwave passive communication components can be seen in current communication products, such as in the base station of 5G communication systems or portable mobile phones. Since the size of the microwave passive component determines the dimension of the wireless communication apparatus, the size reduction of the microwave passive components is always an inevitable trend [1]. The microwave materials with high dielectric constants (ε_r_), high qualify factor (Q × f) values, and small temperature coefficients of the resonant frequency (*τ_f_*) have been extensively developed in the past 20 years to meet the miniaturization requirement of the circuits [2,3,4]. To further reduce the circuit size, bandpass filters and antenna duplexers are suitable to implement on the multilayered ceramic. In order to achieve energy-saving goals and fabricate the multilayer microwave devices, microwave dielectric materials with low sintering temperatures and low-melting-point conductors are needed to co-fire [5]. However, many commercial dielectric materials used for high-frequency device applications have high sintering temperatures; thus, they are not well matched with low-temperature sintering processes [2,3]. Microwave sintering or cold-sintering technologies are also reported to further reduce the sintering temperature of the dielectric materials [6,7]; however, the industrial production is still needed to develop.

In past, three methods—chemical processing [8], low-melting glass addition [9,10], and sintering aid addition [11,12,13,14]—have been commonly used to reduce the sintering temperature of dielectric ceramics. Chemical processing can lower the sintering temperature by altering the particles sizes of the ceramic powders. However, the processing is typically complex and expensive [8]. Low-melting glass addition is a low cost and easy process. Many of the low temperature sintering ceramics using glass additions have already been investigated. However, low-melting glass addition might reduce the microwave dielectric properties of the ceramics, especially in quality factor due to the many defects [9]. Sintering aid addition is also a cost-effective and simple way to reduce the sintering temperature by providing the liquid phase sintering in some specific aids such as ZnO, Bi_2_O_3_, CuO, B_2_O_3_, or V_2_O_5_.

The LaAlO_3_ dielectric ceramic is reported as a phase-stable and low-loss dielectric material, but it has a large negative *τ_f_* value of approximately −45 ppm/°C [15]. The dielectric resonator made by LaAlO_3_ material with such a high negative *τ_f_* produces a large resonant frequency shift when using in the base station operating in a large temperature range. To adjust the temperature coefficient value approaching to zero, an LaAlO_3_ material is combined with different perovskite materials such as SrTiO_3_ or CaTiO_3_ materials with large positive temperature coefficients [16,17,18,19,20,21]. CaTiO_3_ with a large positive *τ_f_* value of ~800 ppm/°C is suitable to tune the *τ_f_* value of the LaAlO_3_ material [18]. With the ratio of 0.95LaAlO_3_–0.05CaTiO_3_, the *τ_f_* value can be tuned to around 0 ppm/°C [20]. However, the sintering temperature needed to densify 0.95LaAlO_3_–0.05CaTiO_3_ ceramics was as high as 1600 °C. It is known that the main effect for liquid phase sintering is that the liquid phase needs to wet the sintered grains and then help grain growth [22]. Since it is cheaper and easier to lower the firing temperature with liquid phase sintering, there is a need to study the effect of liquid phase sintering aids on the sintering and microwave dielectric properties of 0.95LaAlO_3_–0.05CaTiO_3_ ceramics.

In this paper, CuO and B_2_O_3_ are used as the sintering aids due to the appearance of the liquid phase sintering. The effect of additions of 1 wt % CuO, 1 wt % B_2_O_3_, and 0.5 wt % CuO + 0.5 wt % B_2_O_3_ on the sintering behaviors, microstructures, and microwave dielectric properties of 0.95LaAlO_3_–0.05CaTiO_3_ ceramics was investigated.

## 2. Experimental Procedure

The starting materials were La_2_O_3_, Al_2_O_3_, CaCO_3_, and TiO_2_ high-purity oxide powders, which were mixed and ground in distilled water for 12 h in a balling mill with agent balls in relation to the required composition. Mixed powders were dried and then calcined at 1200 °C for 3 h, and then re-milled with different sintering aids for 3 h. In this study, three doping conditions are used to investigate the ability of lowering the sintering temperature, including 1 wt % CuO, 1 wt % B_2_O_3_, and 0.5 wt % CuO + 0.5 wt % B_2_O_3_. After drying, the calcined powder was added with the binder, polyvinyl alcohol (PVA). Pellets with 15 mm diameter that were 5 mm thick were prepared by uniaxial pressing. These pellets were heated at a temperature of 300 °C for 1 h at first to remove the binder. Then, these pellets were sintered at temperatures of 1250–1400 °C for 3 h. Pure 0.95LaAlO_3_–0.05CaTiO_3_ ceramics disk without addition were also prepared through the similar procedure and sintered at temperatures of 1500–1600 °C for 3 h for the comparison.

The bulk densities of the sintered pellets were measured by the Archimedes method. The theoretical densities (D) for the sintered 0.95LaAlO_3_–0.05CaTiO_3_ ceramics can be obtained as
D = (W_1_ + W_2_)/(W_1_/D_1_ + W_2_/D_2_)(1)
where W_1_ and W_2_ are the weight percentages of 0.95LaAlO_3_–0.05CaTiO_3_ ceramics and sintering aids in combination, respectively. D_1_ and D_2_ are the densities of 0.95LaAlO_3_–0.05CaTiO_3_ ceramic and sintering aid in the mixture, respectively. In generally, Equation (1) is only effective when no interaction happened between the dielectric material and the sintering aid.

The crystalline phases of the sintered ceramics were identified by X-ray diffraction patterns (XRD, RIGAKU-2000 X-Ray Diffractometer, Tokyo, Japan). The microstructure of the sintered surface of ceramics was observed by scanning electron microscopy (SEM, JEOL-JSM-6700F, Tokyo, Japan) and energy dispersive spectra (EDS, JEOL-JSM-6700F, Tokyo, Japan).

The dielectric constant ε_r_ and the quality factor Q at microwave frequencies were measured using the Hakki–Coleman dielectric resonator method improved by Courtney [23,24]. The quality factor is further expressed as Q × f, since it is typically constant in the microwave region. A cylindrical dielectric resonator was positioned between two brass plates connected to the measuring system. A HP8757D network analyzer and a HP8350B sweep oscillator were used in the measurement. The temperature coefficient of resonant frequency (*τ_f_*) at the microwave frequency was measured in the temperature range from 20 °C to 80 °C, and estimated by Equation (2),
*τ_f_* = (f_80_ − f_20_)/(60 × f_20_) × 10^6^ (ppm/°C)(2)
where f_20_ and f_80_ are the TE_01δ_ resonant frequency at 20 °C and 80 °C, respectively.

## 3. Results and Discussion

Pure 0.95LaAlO_3_–0.05CaTiO_3_ ceramics without addition were also prepared and used as the standard reference. The sintered samples without sintering aids have relative theoretical densities of 90% and 95% at the sintering temperature of 1500 °C and 1550 °C, respectively. LaAlO_3_ ceramic has approximate dielectric property values of ε_r_ 20, Q × f 65,000 GHz, and *τ_f_* −50 ppm/°C, and CaTiO_3_ ceramic has approximate dielectric properties values of ε_r_ 104, Q × f 3600 GHz, and *τ_f_* 800 ppm/°C. The dielectric properties of the 0.95LaAlO_3_–0.05CaTiO_3_ ceramic is suggested to follow the well-known mixing rules [3]: lnε_r_ = v_1_ lnε_1_ + v_2_ lnε_2_(3)
Q^−1^ = v_1_ Q_1_^−1^ + v_2_ Q_2_^−1^(4)
*τ_f_* = v_1_*τ_f_*_1_ + v_2_*τ_f_*_2_(5)
where v_1_ and v_2_ represent the volume fraction of LaAlO_3_ and CaTiO_3_, respectively. With the ratio of 0.95LaAlO_3_–0.05CaTiO_3_, it exhibits the approximately dielectric property values of ε_r_ 20–22, Q × f 35,000 GHz, and *τ_f_* 1 ppm/°C.

### 3.1. Microstructure

Figure 1 shows the X-ray patterns of 0.95LaAlO_3_–0.05CaTiO_3_ ceramics sintered at different temperatures with the addition of (a) 1 wt % CuO, (b) 1 wt % B_2_O_3_, and (c) 0.5 wt % CuO + 0.5 wt % B_2_O_3_. It is observed from Figure 1 that LaAlO_3_ is the main crystallize phase of the 0.95LaAlO_3_–0.05CaTiO_3_ ceramics. For addition of 1 wt % CuO, as shown in Figure 1a, at the sintering temperature of 1250 °C, LaAlO_3_ and the small amount of CaTiO_3_ are solid-dissolved together, but the secondary phase CaAl_12_O_19_ is still produced. As the sintering temperature rises, the secondary phase gradually disappears. When the sintering temperature is at 1350 °C, the secondary phase completely disappears, and the ceramic is formed as a solid solution.

For additions of 1 wt % B_2_O_3_ and 0.5 wt % CuO + 0.5 wt % B_2_O_3_, the crystallize phases of the 0.95LaAlO_3_–0.05CaTiO_3_ ceramics with different sintering temperatures show similar results. At the sintering temperature of 1250 °C, many secondary phases such as La_5_Ti_5_O_17_, La_4_Ti_9_O_24_, La_2_Ti_2_O_7_, and CaAl_4_O_7_ exist; even LaAlO_3_ and CaTiO_3_ are solid-dissolved together. However, the secondary phase disappears rapidly as the sintering temperature rises. The secondary phase almost completely disappears, and the ceramics have formed a solid solution completely at the sintering temperature of 1350 °C.

Figure 2 shows micrographs of 0.95LaAlO_3_–0.05CaTiO_3_ ceramics sintered at different temperature for 3 h with 1 wt % CuO addition. The melting point of the added CuO is as low as about 1026 °C, which is easy to cause a liquid phase sintering for the 0.95LaAlO_3_–0.05CaTiO_3_ ceramics. Moreover, the liquid phase effect would be enhanced by the eutectic of CuO–Cu_2_O–TiO_2_ (Cu_3_TiO_4_) at 1070 °C [25]. However, due to an inhomogeneous liquid phase distribution, the growth of crystal grains during high temperature sintering is not homogeneous. At sintering temperatures below 1350 °C, many pores exist. As the sintering temperature rises to 1400 °C, the crystal grain is densely arranged on the ceramic surface. However, the internal structure can be found to have a gap, and it is judged that the dense effect is not good. It is reported that the sintering aids would still exist or evaporate after the liquid phase sintering; thus, EDS analysis is used to identify the elements on the surface of the sintered sample [11]. As shown in Figure 2e, Cu atoms are detected as existing on the grains in the surface after the liquid phase sintering.

Figure 3 shows micrographs of 0.95LaAlO_3_–0.05CaTiO_3_ ceramics sintered at different temperatures for 3 h with 1 wt % B_2_O_3_ additions. When the sintering temperature is 1250 °C, the crystal grains dissolve into small particles. As the sintering temperature increases to 1350 °C, the crystal grains grow and gradually become uniform, which leads to more densification. Liquid phase sintering help the growth of crystal grains, and the surface microstructure will be easily covered by the liquid phase [10]. It is resulted that B_2_O_3_ belongs to a low melting point material (about 450 °C) and is a well-known liquid promoter. Moreover, the grain-wetting ability of B_2_O_3_ addition is better than that of CuO addition in 0.95LaAlO_3_–0.05CaTiO_3_ ceramics. Moreover, as shown in Figure 3e, the B atom is also detected as existing in the surface of the grain after the liquid phase sintering.

Figure 4 shows micrographs of 0.95LaAlO_3_–0.05CaTiO_3_ ceramics sintered at different temperatures for 3 h with 0.5 wt % CuO + 0.5 wt % B_2_O_3_ addition. The CuO and B_2_O_3_ added in the experiment are all low-melting materials [13,14]. During the sintering temperatures above 1250 °C, liquid phase sintering is clearly observed, since the crystal grains grow non-homogeneously. When the sintering temperature reaches 1300 °C, the pores have all disappeared. The liquid phase sintering is very obvious, and the complete densification is achieved at the sintering temperature of 1300 °C. It is shown that as the sintering temperature exceeds 1350 °C, an excessive liquid phase is generated, and thus the growth of crystal grains is not homogeneous, and the secondary recrystallization is formed, which would affect the performance of the dielectric characteristics. Secondary recrystallization typically occurs when continuous grain growth is inhibited by the present of impurities or the liquid phase, as shown in Figure 4d. It is also identified that Cu and B atoms are both detected as existing in the surface of the grain after liquid phase sintering, as shown in Figure 4e.

Figure 5 shows the grain size of 0.95LaAlO_3_–0.05CaTiO_3_ ceramics as the functions of sintering temperature with different sintering aids. Typically, with the increase of sintering temperature, the surface grain size tends to increase, indicating that high temperature sintering can promote grain growth. It is indicated that the grain size of the ceramics with the addition of CuO is about 0.6–1.2 μm, which is larger than others. However, many pores exist in the sintered ceramics. With B_2_O_3_ addition, although the grain is close to dense, the grain growth is still slow due to a small grain size of about 0.6–0.8 μm. With 0.5 wt % CuO + 0.5 wt % B_2_O_3_ addition, the surface is also dense, but the grain growth is increased slowly due to many small grain sizes. In the excessive liquid phase at a sintering temperature of 1400 °C, the surface grain size could not be estimated.

Figure 6 shows a shrinkage ratio of 0.95LaAlO_3_–0.05CaTiO_3_ ceramics sintered at different temperatures with different additions. The shrinkage ratio is defined as the difference of the sample diameter before and after sintering over the sample diameter before sintering. It is found that the shrinkage ratios of 0.95LaAlO_3_–0.05CaTiO_3_ ceramics are around 4–6%, 12–17% and 13–18% for the additions of 1 wt % CuO, 1 wt % B_2_O_3_, and 0.5 wt % CuO + 0.5 wt % B_2_O_3_, respectively. Although the grain sizes of the sintered sample added with 1 wt % CuO are larger than those added with 1 wt % B_2_O_3_ and 0.5 wt % CuO + 0.5 wt % B_2_O_3_, the shrinkage ratios are the worst due to non-homogeneous grain growth and the existence of many pores. At 1350 °C, the shrinkage ratio of the sintered sample reaches the maximum relative value, showing that B_2_O_3_ addition helps the 0.95LaAlO_3_–0.05CaTiO_3_ ceramics to shrink and be compact.

For density measurements, the 0.95LaAlO_3_–0.05CaTiO_3_ ceramics without sintering aid addition were also observed as the standard. The 0.95LaAlO_3_–0.05CaTiO_3_ ceramic added with 0.5 wt % CuO + 0.5 wt % B_2_O_3_ can obtain the measured density of 6.01 g/cm^3^, which is a relative theoretical density of 90%, at the sintering temperature of 1250 °C, and the measured density of 6.05 g/cm^3^, which is a relative theoretical density of 94%, at the sintering temperature above 1300 °C. The 0.95LaAlO_3_–0.05CaTiO_3_ ceramics added with 1 wt % B_2_O_3_ obtain relative theoretical densities of around 80–90% at the sintering temperature of 1300–1400 °C. However, the 0.95LaAlO_3_–0.05CaTiO_3_ ceramic added with 1 wt % CuO have relative theoretical densities less than 90% at a sintering temperature of 1400 °C, which is the same finding as that from the shrinkage. It shows that the addition of B_2_O_3_ combined with CuO helps the compaction of 0.95LaAlO_3_–0.05CaTiO_3_ ceramics and reduces the sintering temperature by nearly 250 °C, as compared to those without sintering aids.

### 3.2. Dielectric Properties

Figure 7 indicates dielectric constants (ε_r_) of 0.95LaAlO_3_–0.05CaTiO_3_ ceramics as the functions of sintering temperature with different additions. In the sintering range, 0.95LaAlO_3_–0.05CaTiO_3_ ceramics added with 0.5 wt % CuO + 0.5 wt % B_2_O_3_ have higher dielectric constants of around 19–20.5, which resulted from the full condensation. 0.95LaAlO_3_–0.05CaTiO_3_ ceramics added with 1 wt % B_2_O_3_ have dielectric constants of around 15.8–18.2 due to the lower densification and the excessive secondary phase formation at lower sintering temperatures. The 0.95LaAlO_3_–0.05CaTiO_3_ ceramics added with 1 wt % CuO have low dielectric constants due to the low density and the excess pores with air phase (ε_r_ = 1) [2].

Figure 8 indicates the quality factor Q × f values of 0.95LaAlO_3_–0.05CaTiO_3_ ceramics as the functions of the sintering temperature with different additions. It is known that the microwave dielectric loss is simultaneously affected by many factors, which are mainly caused not only by densification, pores, and grain sizes/boundaries, and but also by the lattice secondary phases and vibration modes [8]. It seems that the relative theoretical density and pores are important factors in controlling the dielectric loss. As shown in Figure 8, the 0.95LaAlO_3_–0.05CaTiO_3_ ceramic added with 0.5 wt % CuO + 0.5 wt % B_2_O_3_ has a higher Q × f value of around 22,500 GHz at the sintering temperature of 1300 °C, which is near the Q × f value of the 0.95LaAlO_3_–0.05CaTiO_3_ ceramic without sintered aid sintered at 1550 °C. After the sintering temperature exceeds 1350 °C, the Q × f value decreases, which is presumed to be caused by the excessive liquid phase of the ceramic. The 0.95LaAlO_3_–0.05CaTiO_3_ ceramics added with 1 wt % B_2_O_3_ have Q × f values of around 1000–12,000 GHz, which is affected by the low densification [3]. The 0.95LaAlO_3_–0.05CaTiO_3_ ceramics added with 1 wt % CuO have low densities and contain too much pores, resulting in the worst performance of Q × f values.

Figure 9 indicates *τ_f_* values of 0.95LaAlO_3_–0.05CaTiO_3_ ceramics as the functions of sintering temperature with different additions. The *τ_f_* values of 0.95LaAlO_3_–0.05CaTiO_3_ ceramics added with 1 wt % CuO and 1 wt % B_2_O_3_ have a negative value and a positive value, respectively. It is found that the 0.95LaAlO_3_–0.05CaTiO_3_ ceramics added with 0.5 wt % CuO + 0.5 wt % B_2_O_3_ have an acceptable temperature coefficient of resonant frequency (*τ_f_*) approaching zero value. Generally, the temperature coefficient of resonant frequency (*τ_f_*) is dependent on the composition and the existing phase in the ceramics [10]. In this study, the sintered ceramic with CuO addition has a negative temperature coefficient, while that with B_2_O_3_ addition has a positive temperature coefficient. Since the ratio of CaTiO_3_ to LaAlO_3_ is also appropriate, a system with a temperature coefficient approaching zero can be obtained.

## 4. Conclusions

LaAlO_3_ is a quite potential microwave dielectric material with a dielectric constant (ε_r_) of 23, a quality factor (Q × f) value of 65,000 GHz, and a negative temperature coefficient of temperature (*τ_f_*) of −44 ppm/°C. For mixing CaTiO_3_ ceramic as compensation for the temperature coefficient of resonant frequency, 0.95LaAlO_3_–0.05CaTiO_3_ ceramic is used in this study. In order to reduce the sintering temperature, CuO and B_2_O_3_ sintering aids for liquid phase sintering are added to the 0.95LaAlO_3_–0.05CaTiO_3_ ceramic to help grain growth, to increase the density, as well as to obtain acceptable quality factors of the 0.95LaAlO_3_–0.05CaTiO_3_ ceramics.

The effects of the liquid phase sintering aids added to the 0.95LaAlO_3_–0.05CaTiO_3_ ceramic are summarized as follows.
For the addition of 1 wt % CuO to the 0.95LaAlO_3_–0.05CaTiO_3_ ceramics, the effect is the worst. When the sintering temperature is 1400 °C, the dielectric constant (ε_r_) is about 16, the highest quality factor (Q × f) value is about 7500 GHz, and the temperature coefficient of resonant frequency (*τ_f_*) is about −11 ppm/°C.For the addition of 1 wt % B_2_O_3_ to the 0.95LaAlO_3_–0.05CaTiO_3_ ceramics, the sintering temperature is lowered to 1400 °C, the electrical constant (ε_r_) is about 18, the highest quality factor (Q × f) value is about 10,000 GHz and the temperature coefficient of resonant frequency (*τ_f_*) is about −5 ppm/°C.For the addition of 0.5 wt % CuO + 0.5 wt % B_2_O_3_ to the 0.95LaAlO_3_–0.05CaTiO_3_ ceramics, its sintering temperature can be reduced to 1300 °C (about 250 °C reduction, as compared to the same relative theoretic density of the 0.95LaAlO_3_–0.05CaTiO_3_ ceramic without a sintering aid), the dielectric constant (ε_r_) is about 21, the highest quality factor (Q × f) value is 22,500 GHz, and the temperature coefficient of the resonant frequency (*τ_f_*) is about −3 ppm/°C.

## Figures and Tables

**Figure 1 materials-12-04187-f001:**
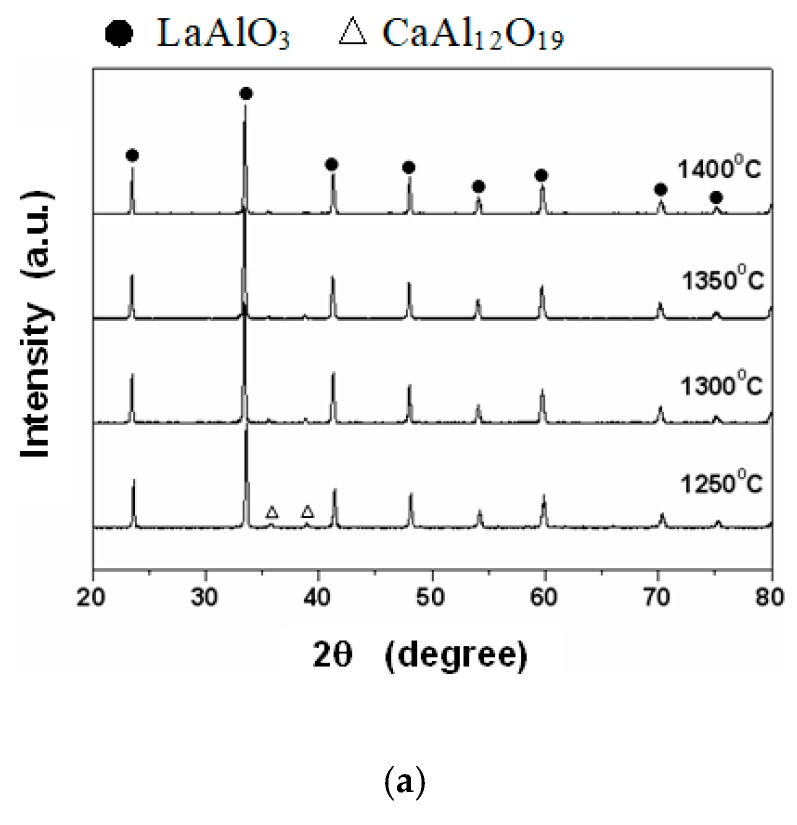
The X-ray patterns of 0.95LaAlO_3_–0.05CaTiO_3_ ceramics sintered at different temperatures with the addition of (**a**) 1 wt % CuO, (**b**) 1 wt % B_2_O_3_, and (**c**) 0.5 wt % CuO + 0.5 wt % B_2_O_3_.

**Figure 2 materials-12-04187-f002:**
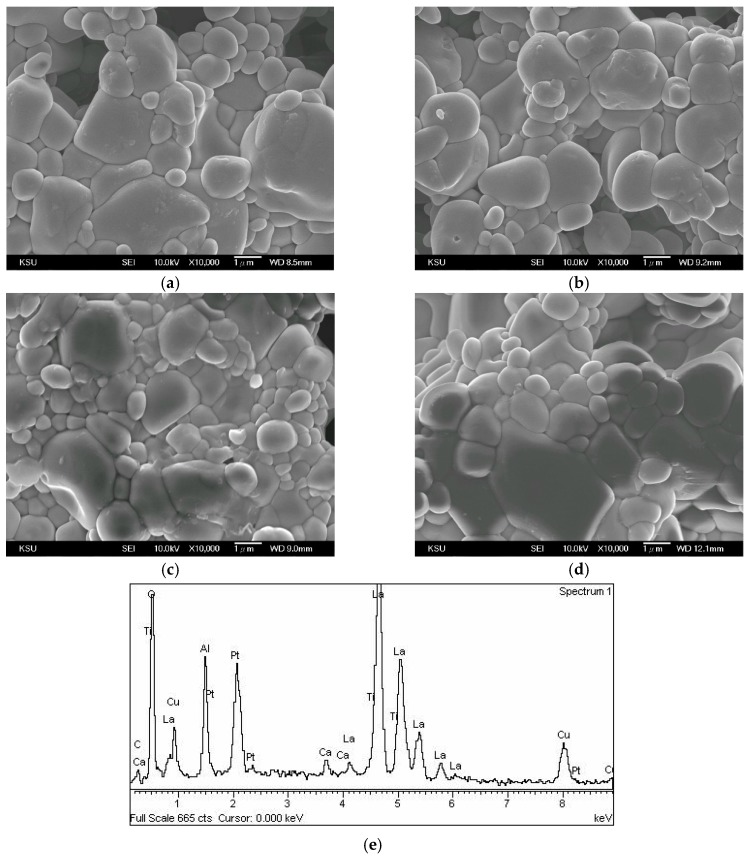
Micrographs of 0.95LaAlO_3_–0.05CaTiO_3_ ceramics sintered at different temperatures for 3 h with 1 wt % CuO addition at (**a**) 1250 °C, (**b**) 1300 °C, (**c**) 1350 °C, (**d**) 1400 °C, and (**e**) energy dispersive spectra (EDS) analysis of surface of the sintered ceramic (**c**).

**Figure 3 materials-12-04187-f003:**
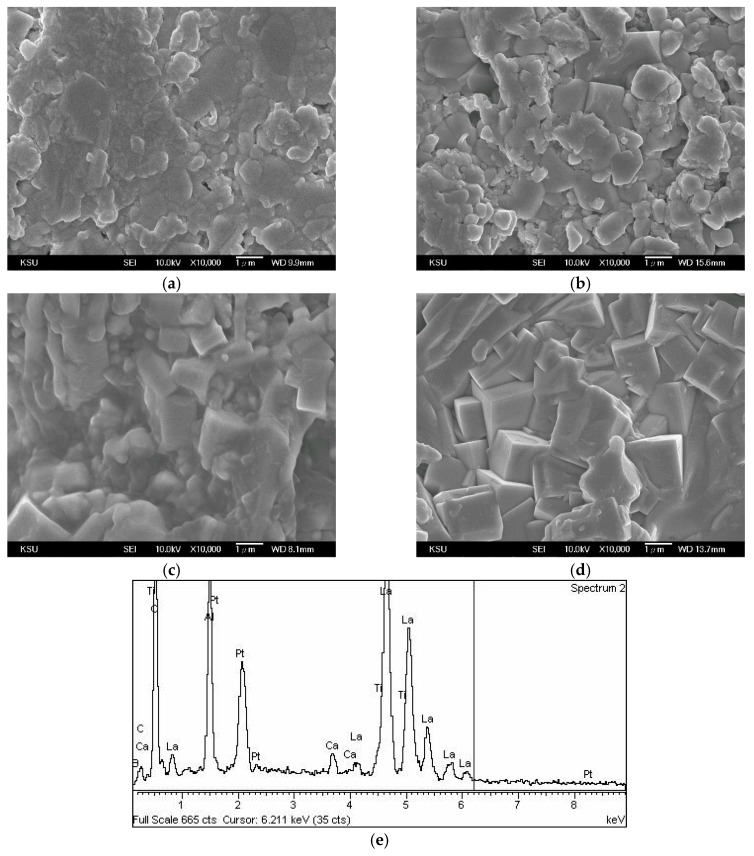
Micrographs of 0.95LaAlO_3_–0.05CaTiO_3_ ceramics sintered at different temperature for 3 h with 1 wt % B_2_O_3_ addition at (**a**) 1250 °C, (**b**) 1300 °C, (**c**) 1350 °C, (**d**) 1400 °C, and (**e**) EDS analysis of the surface of the sintered ceramic (**c**).

**Figure 4 materials-12-04187-f004:**
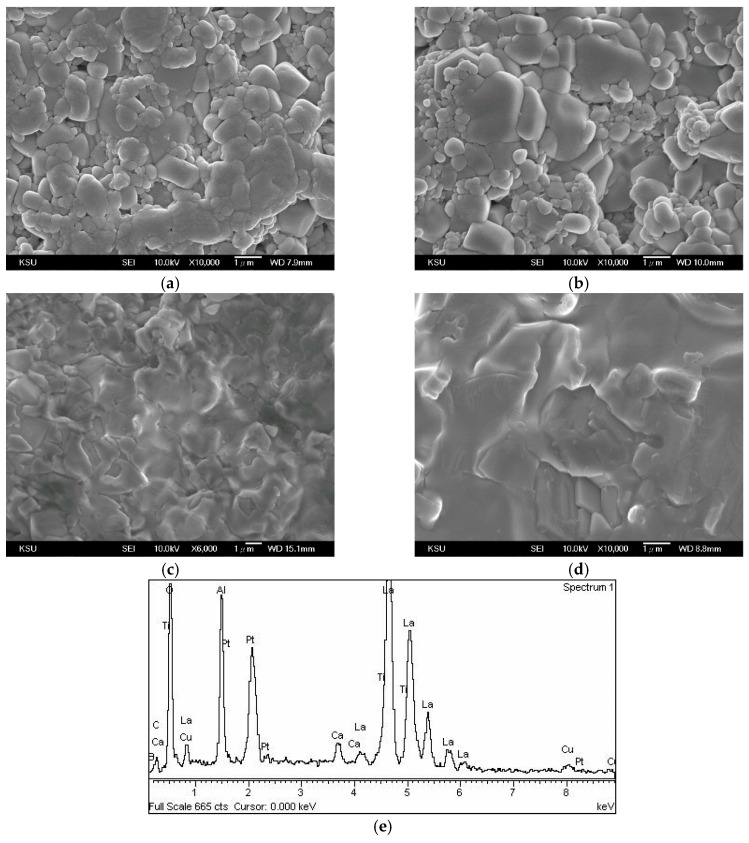
Micrographs of 0.95LaAlO_3_–0.05CaTiO_3_ ceramics sintered at different temperature for 3 h with 0.5 wt % CuO + 0.5 wt % B_2_O_3_ addition at (**a**) 1250 °C, (**b**) 1300 °C, (**c**) 1350 °C, (**d**) 1400 °C, and (**e**) EDS analysis of surface of the sintered ceramic (**c**).

**Figure 5 materials-12-04187-f005:**
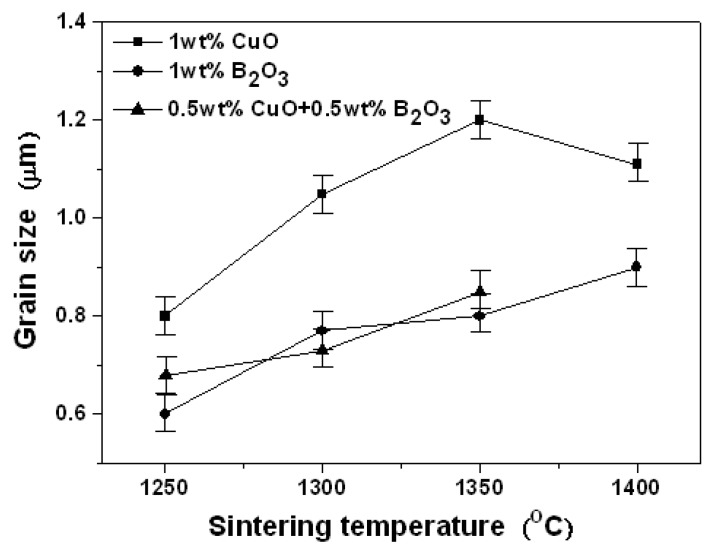
The grain size of 0.95LaAlO_3_–0.05CaTiO_3_ ceramics as a function of sintering temperature.

**Figure 6 materials-12-04187-f006:**
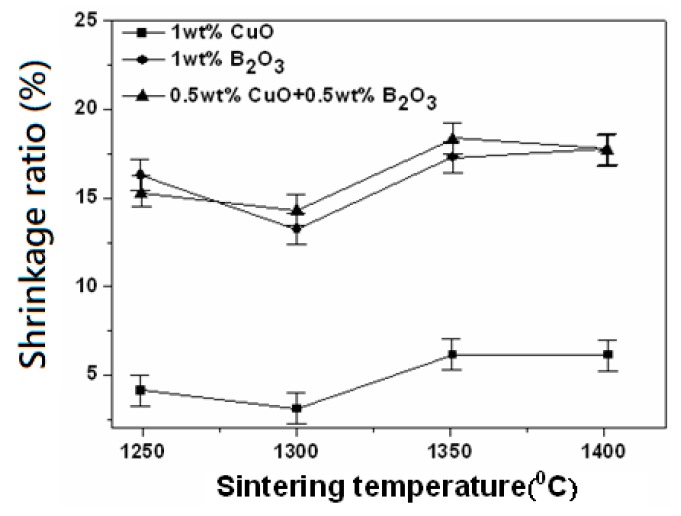
Shrinkage of 0.95LaAlO_3_–0.05CaTiO_3_ ceramics sintered at different temperatures with different additions.

**Figure 7 materials-12-04187-f007:**
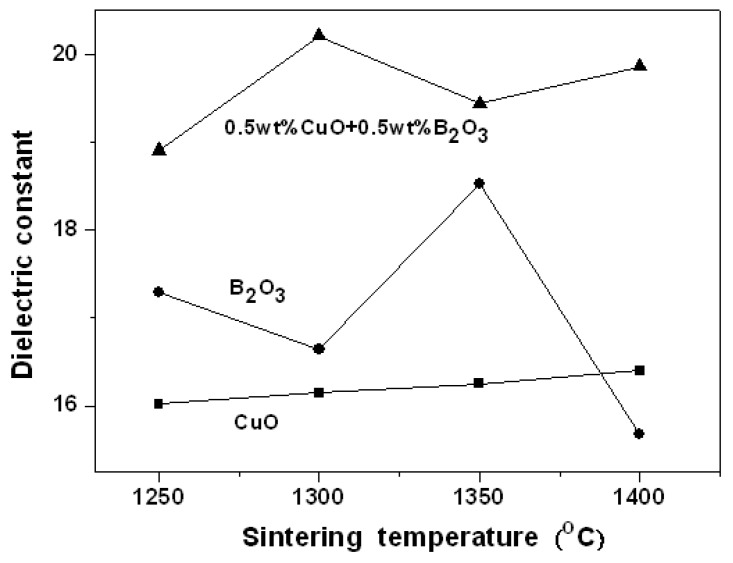
Dielectric constants of 0.95LaAlO_3_–0.05CaTiO_3_ ceramics as the functions of sintering temperatures with different additions.

**Figure 8 materials-12-04187-f008:**
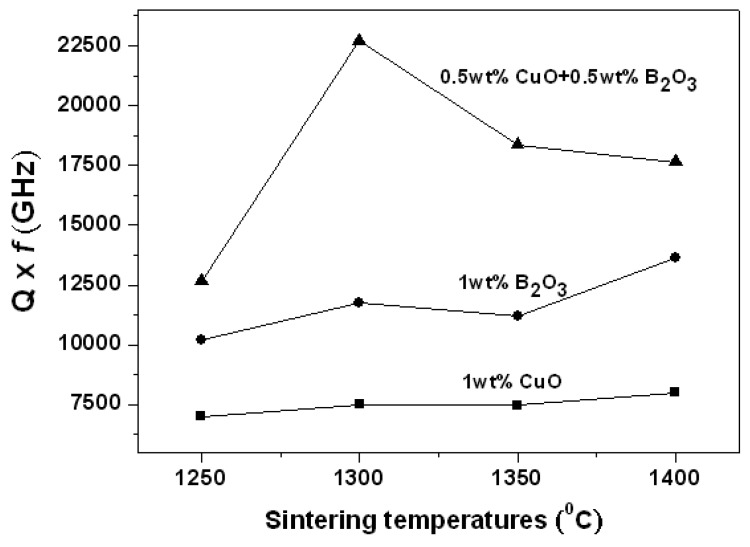
Q × f values of 0.95LaAlO_3_–0.05CaTiO_3_ ceramics as the functions of sintering temperature with different additions.

**Figure 9 materials-12-04187-f009:**
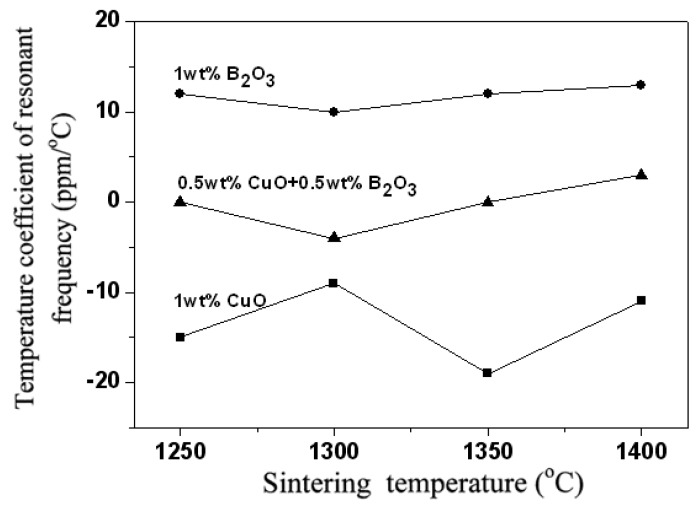
*τ_f_* values of 0.95LaAlO_3_–0.05CaTiO_3_ ceramics as the functions of sintering temperature with different additions.

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
