# Peer review of "Sintering Behaviors, Microstructure, and Microwave Dielectric Properties of CaTiO3–LaAlO3 Ceramics Using CuO/B2O3 Additions"

_materials, 2019, doi:10.3390/ma12244187_

Round 1

Reviewer 1 Report

Time unit, hr or hours should be changed into h. Line #51 - 53: The sentence should be clearly rewritten. Line #181:  , which is large than others → which is larger than others Line #195: the shrinkage ratio of is worst →  the shrinkage ratio is worst

Author Response

Reply for Reviewer 1's Comments

Time unit, hr or hours should be changed into h. Line #51 - 53: The sentence should be clearly rewritten. Line #181: , which is large than others → which is larger than others Line #195: the shrinkage ratio of is worst → the shrinkage ratio is worst

Reply:

These comments are well received. We have corrected all the errors suggested by the reviewer 1 and marked the corrections using yellow color in the revised word file.

Reviewer 2 Report

The manuscript shows results about the effect of CuO/B2O3 additions on the sintering behaviors, microstructures and microwave dielectric properties of 0.95LaAlO3-0.05CaTiO3 ceramics. The authors should make investigations (TEM, EDS or other) where are incorporated the Cu or B atoms? How the authors know that liquid phase sintering occured? A thorough TEM investigation on grain boundaries should be performed. Even an  EDS on SEM would be useful to show the distribution of Cu or B elements on powders or sintered samples. This information is completely missing.

Author Response

The comments from the reviewer are very constructive to our paper. The paper is revised in accordance with the reviewer’s comments and has marked the corrections using yellow color in the revised word file. The attached are the revision and the reply letter for the comments. There are listed as follows:

Answers for Reviewer 2's Comments

The manuscript shows results about the effect of CuO/B2O3 additions on the sintering behaviors, microstructures and microwave dielectric properties of 0.95LaAlO3-0.05CaTiO3 ceramics. The authors should make investigations (TEM, EDS or other) where are incorporated the Cu or B atoms? How the authors know that liquid phase sintering occured? A thorough TEM investigation on grain boundaries should be performed. Even an EDS on SEM would be useful to show the distribution of Cu or B elements on powders or sintered samples. This information is completely missing.

Reply:

These comments are well received. We have performed EDS on SEM to identify the distribution of Cu or B elements on the sintered samples, and marked the corrections using yellow color in the revised word file.

As discussed in many reports, liquid phase sintering have three steps, including: 1. Rearrangement, 2. solution-reprecipitation, and 3. solid state. The main effect for liquid phase sintering is that the liquid phase must wet the sintered grains and then help grain growth [24]. The liquid phase sintering can be typically observed by SEM.

Typically, although the liquid phase sintering of ceramic can help grain growth, the grain size might be not uniform since the liquid phase is not distributed uniformly. In this study, the melting point of the added CuO is as low as about 1026°C, which is easy to cause a liquid phase sintering for the 0.95LaAlO3-0.05CaTiO3 ceramics. Moreover, the liquid phase effect would be enhanced by eutectic of CuO–Cu2O–TiO2 (Cu3TiO4) at 1070°C. In another, B2O3 belongs to a low melting point material (about 450 °C) and is a well-known liquid promoter. It is found the grain-wetting ability of B2O3 addition is better than that of CuO addition in 0.95LaAlO3-0.05CaTiO3 ceramics. In our results, as shown in Figure 2, 3 and 4, liquid phase sintering are clearly observed.

It is reported that the sintering aids would be still existed or evaporated after the liquid phase sintering, thus EDS analysis is used to identify the elements on the surface of the sintered sample. We have performed EDS on SEM to identify the distribution of Cu or B elements on the sintered samples, and identified that Cu and B atoms are both detected and existed in the surface of the grain after the liquid phase sintering, as shown in Figure 2 (e), 3(e) and 4 (e).

[24] Marison, J. E.; Hsueh, C. H.; Evans, A. G. Liquid phase sintering of ceramics, J. Am. Ceram. Soc, 1987, 70, 708 – 713.

Round 2

Reviewer 2 Report

The  manuscript was considerably improved.

Author Response

Thanks